# Pseudo-Bayesian Approach for Robust Mode Detection and Extraction Based on the STFT

**DOI:** 10.3390/s23010085

**Published:** 2022-12-22

**Authors:** Quentin Legros, Dominique Fourer

**Affiliations:** 1LTCI, Télécom Paris, 91120 Palaiseau, France; 2IBISC, Université d’Évry/Paris-Saclay, 91080 Évry-Courcouronnes, France

**Keywords:** time-frequency, nonstationary component estimation, robust divergences, variational approximation, hypothesis test, assumed density filtering, synchrosqueezing

## Abstract

This paper addresses the problem of disentangling nonoverlapping multicomponent signals from their observation being possibly contaminated by external additive noise. We aim to extract and to retrieve the elementary components (also called modes) present in an observed nonstationary mixture signal. To this end, we propose a new pseudo-Bayesian algorithm to perform the estimation of the instantaneous frequency of the signal modes from their time-frequency representation. In a second time, a detection algorithm is developed to restrict the time region where each signal component behaves, to enhance quality of the reconstructed signal. We finally deal with the presence of noise in the vicinity of the estimated instantaneous frequency by introducing a new reconstruction approach relying on nonbinary band-pass synthesis filters. We validate our methods by comparing their reconstruction performance to state-of-the-art approaches through several experiments involving both synthetic and real-world data under different experimental conditions.

## 1. Introduction

Fast and reliable disentangling of natural signals is a necessary step for a variety of applications including biomedicine, audio, seismic applications, and radar [1,2]. Such complex signals are often generated by physical systems that can be modeled as a superimposition of Amplitude-and Frequency-Modulated (AM-FM) waves (or modes), and referred to as MultiComponent Signal (MCS).

Although extracting the signal components can be challenging depending on the signal nature or its acquisition, a large number of techniques have been developed to tackle those limitations. From the different existing approaches, expanding MCSs is often completed by computing a Time-Frequency Representation (TFR) (resp. time-scale representation) by using popular methods such as the Short-Time Fourier Transform (STFT) or the Continuous Wavelet Transform (CWT) [3], which can reveal curves (also called ridges), associated with each signal mode in the time-frequency plane. Estimating the position of the ridges approximates the Instantaneous Frequency (IF) of each mode, which can then be extracted and reconstructed through an adapted filtering method [4]. Knowing the ridges of a signal enables a large variety of advanced applications such as signal enhancement and denoising, source separation, or information extraction [5,6,7]. The main motivation of this work is to enhance signal analysis through MCS modeling. Knowing the main components of a signal is almost always sufficient to extract relevant information and to estimate physics-related parameters as proposed in [7]. Although disentangling a signal can lead to a better understanding of the observed phenomena, it also provides the opportunity to remove undesired noise and to improve its readability.

Several approaches have been proposed during the last decades to extract the modes of a MCS. Data-driven methods such as Empirical Mode Decomposition (EMD) [8] or Singular Spectrum Analysis (SSA) [9] allow us to decompose an arbitrary time series into a set of intrinsic mode functions. Although these techniques can obtain promising results in a large variety of real-world applications, they suffer from a lack of robustness when dealing with AM-FM modes, or in the presence of external noise, involving the need of adapting the original approach [10]. On the other hand, other methods perform ridge estimation by interpolating the curves along the time axis by using the detected local maximums [11]. This enables intensive studies, especially in the field of audio processing, for which partial tracking algorithms have been previously proposed for dealing with harmonic signal components made of a fundamental frequency and its integer multiples [12]. Because the estimation performance of the ridges depends on the presence of interfering components, efforts are made to improve the accuracy and robustness of the methods [13]. Existing work improves the readability of the computed TFR using the Synchrosqueezing transform (SST) [4,14,15], possibly combined with a demodulation technique [16] to obtain sharpened and reversible TFR for improving modes retrieval. A filtering method based on spectrogram zeros was proposed in [17] for extracting the signal components in the time-frequency plane. Even though this approach can deal with the presence of noise, it becomes limited in extreme scenarios involving low Signal-to-Noise Ratio (SNR). Although the presence of noise can be considered as the main challenge for estimating MCS modes, a wide range of methods in the literature are deterministic and often neglect the presence of outliers. Estimating modes of MCS remains an interesting problem depending on the quality of the observations and of the acquisition conditions that can traduce incomplete data or the presence of external noise. Nonetheless, the presence of noise is not the unique reason of mismatch between a postulated observation model and observed data. For instance, some approaches have been devoted to identify the effective time support of a signal to achieve the mode reconstruction based on TFR ridge detection. For example in [18], an indicator term is plugged in a classical MCS observation model to inform on the presence or lack of a ridge. The algorithm estimates the noise spectrum through multipass filtering, before detecting the signal peaks associated with signal components by using a hypothesis test. A similar model was proposed in [19], in which the spectrogram is first segmented to highlight modes birth (start) and death (end), before performing IF estimation by using a peak tracking method. Bayesian methods [5,6], which rely on the computation of a posterior probability function to infer estimates, can be easily adapted for estimating model parameters from noisy observations. Although related approaches can significantly improve the estimation performances of MCS modes in challenging scenarios, they remain computationally expensive.

In this work, we address the problem of reconstructing MCS from observations corrupted by external noise. Although it is common to focus on a specific step of the whole reconstruction process, we propose three different methods aiming, respectively, to perform robust estimation of the ridges position in a TFR to mitigate the noise effect during the signal reconstruction. To this aim, we estimate for the first time the components’ IF from its TFR by using a Bayesian framework. A simple observation model is postulated to ensure computational tractability of further inference process. Although the lack of generality of the observation model avoids proper estimation of the model parameters in the case of model mismatch, an adapted estimation strategy is used to enhance the robustness of the method. Moreover, a novel objective function is introduced to regularize the estimation process according to both the presence of multiple ridges or a strong noise level involving possible model mismatch. This generalizes and extends our previous contribution [20], which only addressed one of these two constraints separately by respectively minimizing a divergence enforcing either the robustness or the mass-covering character of the variational approximation [21]. More precisely, an Alpha-Beta (AB) variational objective [21,22] is used to obtain a more general control over the inference process.

For a second time, we propose a new detection algorithm that uses an alternative observation model to predict, for each time instant, whether the signal components are present or not. Because the proposed estimation algorithm assumes that information is present at each time instant for each signal component, its output needs to be processed to perform an efficient reconstruction of the MCS. In comparison to existing methods, such as [18,19], our approach uses specific mathematical derivations which lead to a lower computation cost while remaining robust in the presence of strong noise. The third contribution of this work is related to the mode reconstruction. In the presence of noise, classical methods tend to denoise the signal in the time-frequency plane by performing reconstruction from the frequency bands associated with the estimated ridges. Here, we propose a reconstruction strategy, reducing the noise-related energy used to synthesize the MCS. The main contributions of the paper can thus be summarized as follows.
A novel pseudo-Bayesian estimation algorithm for ridge extraction based on an alternative variational objective allowing for efficient regularization is discussed.A new fast and reliable detection algorithm for determining the time support of each of the MCS frequency component is discussed.A new denoising strategy for signal reconstruction, mitigating the noise content present in frequency bands used for signal synthesis is discussed.

The remainder of the paper is organized as follows. Section 2 introduces the problem addressed in this work. An extended Pseudo-Bayesian (PB) estimation method is presented by using a new AB variational objective in Section 3. In Section 4, we introduce the detection algorithm used to postprocess the ridge estimation, and the denoising strategy for improving the signal reconstruction performance is detailed in Section 5. Results of experiments conducted with both synthetic and real-world signals are presented in Section 6 before the conclusion is reported in Section 7.

## 2. Problem Statement

In the remainder of this paper, we assume a signal *x* defined as a mixture of *K* superimposed AM-FM components expressed as:(1)$$x\left(t\right)=\sum_{k=1}^{K}x_{k}\left(t\right),\;\mathrm{with}\;x_{k}\left(t\right)=a_{k}\left(t\right)\;e^{j\phi_{k}\left(t\right)},$$
where $a_{k}\left(t\right)$ (resp. $\phi_{k}\left(t\right)$) denotes the time-varying amplitude (resp. phase) of the *k*-th component. Each signal component can be characterized by its ridge located at the IF. By using the STFT synthesis formula given by Equation (Equation 35) or Equation (Equation 38) for its synchrosqueezed version, an accurate approximation of the reconstructed component can be obtained through band-pass filtering on *x*, in order to only preserve the energy at the IF vicinity. Indeed, when the frequency is slowly varying, or when the spread of the window is sufficiently small, the STFT of the *k*-th component can be approximated according to Equtaion (Equation 34) as
(2)$$F_{k}^{h}\left(t,\omega \right)\approx x_{k}\left(t\right)F_{h}\left(\frac{d\phi_{k}}{dt}\left(t\right)-\omega \right)\;e^{-j\omega t},$$
where the IF of each component is by definition the derivative of the phase with respect to time $\frac{d\phi_{k}}{dt}\left(t\right)$. A good approximate of the IF of each component can be obtained by considering the position of the local maxima in the TFR.

In this study, we assume that the ridges associated with the MCS are separable and do not overlap in the time-frequency plane. From such an assumption, it can be stated that the MCS can be approximated by using both Equations (Equation 35) and (Equation 2) by restricting the integrating region to the vicinity of the components IF [23] when $t_{0}=0$, as
(3)$$x_{k}\left(t\right)\approx \frac{1}{h{\left(0\right)}^{*}}\int_{|\omega -\frac{d\phi_{k}}{dt}\left(t\right)|<\epsilon }F_{x}^{h}\left(t,\omega \right)\;e^{j\omega t}\;\frac{d\omega }{2\pi },$$
where ϵ is an arbitrary small threshold depending on the spread of the analysis window. Note that Equation (Equation 38) should be considered when using the synchrosqueezed STFT. In practice, only a noisy version *y* of *x* is observed, such that y=x+e, with *e* standing for an arbitrary additive noise. The aim of this work is to provide a fast and robust estimation framework for estimating the ridges and extracting the modes of *x* from the noisy observations *y*. The main challenge imposed by this problem comes from the distribution of the noise that spreads along the time-frequency plane [24,25]. In the remainder of this study, we consider that the signal has been discretized by using a sampling period $T_{s}$. The previously defined transforms are computed by using the rectangle approximation method. We consider the STFT $F_{y}^{h}\left[n,m\right]\;\approx \;F_{y}^{h}\left(nT_{s},2\pi \frac{m}{MT_{s}}\right)$ at time index n∈{0,1,⋯,N−1} and frequency bin m∈{0,1,⋯,M−1}, with *M* (resp. *N*) being the number of frequency bins (resp. time indices). We use a Gaussian analysis window $h\left(t\right)=\frac{1}{\sqrt{2\pi }T}\;e^{-\frac{t^{2}}{2T^{2}}}$ for which the Fourier transform can be expressed as
(4)$$F_{h}\left(\omega \right)=\;e^{-\frac{\omega^{2}T^{2}}{2}}.$$

The time-spread parameter of *h* considered after the discretization process is defined as $L=\frac{T}{T_{s}}$.

## 3. Pseudo-Bayesian Analysis

In this section, we present the basis of our proposed Pseudo-Bayesian (PB) algorithm for estimating the IFs of the MCS modes through the ridges positions in the 2D time-frequency plane.

### 3.1. Observation Model

Let S be the M×N spectrogram $S_{y}^{h}\;=\;{\left|F_{y}^{h}\right|}^{2}$, such that $s_{n}={\left[S\right]}_{n,:}={\left[s_{n,0},\ldots ,s_{n,M-1}\right]}^{⊤}$. The observations *y* are modeled through the columns $s_{n}$ of S as follows, assuming the presence of only one component and neglecting other external sources. We define our observation model as
(5)$$s_{n,m}|{\bar{m}}_{n}∼g\left(m-{\bar{m}}_{n}\right),$$
where $g\left(m\right)=\frac{2\sqrt{\pi }L}{M}\;e^{-{\left(\frac{2\pi mL}{M}\right)}^{2}}$ is the normalized and discretized squared modulus of the Fourier transform of *h*, and ${\bar{m}}_{n}$ is the position of the ridge associated with the signal IF at time *n*. Although the analysis window width involves correlation between consecutive spectrogram frames, we assume independence between the successive $s_{n}$. Because the method proposed in this paper aims to sequentially perform estimation of the IF at each time instant, the spatial correlation along the time axis is left to the prior distribution model. The joint likelihood function can thus be computed as
(6)$$p\left(s_{n}|{\bar{m}}_{n}\right)=\prod_{m=0}^{M-1}p\left(s_{n,m}|{\bar{m}}_{n}\right).$$

### 3.2. Variational Objective

Even though the observation model in Equation (Equation 5) allows a fast and simple estimation process, its lack of accuracy avoids achieving a satisfying estimation performance in the presence of either multiple components or of external spurious noise. The presence of outliers will involve a model mismatch, avoiding approaches based on Maximum Likelihood Estimation (MLE) to perform in a satisfactory manner. To circumvent the lack of generality imposed by the model, alternative divergences have been used to infer estimates with modified variational objectives. More precisely, performing MLE is equivalent to minimizing the Kullback-Leibler Divergence (KLD) between the data distribution and the postulated model. Note that in practice, the empirical data distribution is used to approximate the unknown true distribution of the data. By replacing the KLD by another divergence, the final objective is correspondingly updated to account for the properties of the new functional. In Bayesian inference, the random variable ${\bar{m}}_{n}$ is assigned a prior distribution $p({\bar{m}}_{n})$, and the posterior distribution $p({\bar{m}}_{n}|s_{n})$ is derived by using Bayes’ theorem as
(7)$$p\left({\bar{m}}_{n}|s_{n}\right)=\frac{p\left(s_{n}|{\bar{m}}_{n}\right)p\left({\bar{m}}_{n}\right)}{p(s_{n})}.$$

Nonetheless, the computation of the posterior probability is generally intractable due to the evidence $p(s_{n})$, because it requires integration over all configurations of the hidden variables. It is, however, possible to approximate this distribution by resorting to variational methods without having to compute the probability of the observation. Variational methods aim to compute the closest approximate distribution to the true posterior distribution. In practice, this is performed by maximizing the Evidence Lower-Bound (ELBO), which is a lower bound of the posterior distribution [26]. Equivalently, it remains to solve
(8)$$\begin{array}{cc} \underset{q({\bar{m}}_{n})\in P}{arg min}\;E_{q({\bar{m}}_{n})} & \left[D_{\mathrm{KL}}(\hat{p}\left(s_{n}\right)||p\left(s_{n}|{\bar{m}}_{n}\right))\right]\\ + & \frac{1}{M}D_{\mathrm{KL}}(q\left({\bar{m}}_{n}\right)||p\left({\bar{m}}_{n}\right)) \end{array},$$
with P the set of all probability distributions, chosen to be analytically tractable in practice, $E_{q({\bar{m}}_{n})}\left[\cdot \right]$ the expectation with respect to $q({\bar{m}}_{n})$ and with the KLD defined as
(9)$$D_{\mathrm{KL}}(\hat{p}\left(s\right)||p\left(s|{\bar{m}}_{n}\right))=\int \hat{p}\left(s\right)\log\left(\frac{\hat{p}\left(s\right)}{p(s|{\bar{m}}_{n})}\right)ds.$$

Although the first term in Equtaion (Equation 8) is the expectation of the Kullback–Leibler Cross Entropy (CE) corresponding to the expected likelihood, the second term constrains the solution to be close to the prior distribution. The variational objective can be modified to account for the lack of accuracy of the postulated model by replacing the KLD used in the first term of Equtaion (Equation 8) by another divergence. In [20], both the α-Divergence (α-D) and the β-Divergence (β-D) were proposed, providing, respectively, robustness of the estimates and a control over the mass-covering behavior of the variational approximation.

### 3.3. Estimation Strategy

We use an iterative estimation strategy in the PB algorithm, where each array $s_{n}$ is processed sequentially. At each time instant *n*, a point estimate for the IF is computed through Minimum Mean Squared Error (MMSE) from the posterior distribution. The prior model used to compute the posterior distribution is chosen to constrain the IFs estimates to slowly evolve along the time axis. For that purpose, a Gaussian Random Walk (GRW) prior model is used to propagate the information as
(10)$$p\left({\bar{m}}_{n+1}\right)\propto N\left({\hat{m}}_{n},\sigma_{rw}^{2}\right){\hat{N}}_{n},$$
where the Gaussian random walk $N({\hat{m}}_{n},\sigma_{rw}^{2})$ models the allowed variation of the IF between successive time instants through $\sigma_{rw}^{2}$. The Gaussian density ${\hat{N}}_{n}$ is an approximation of the posterior distribution at time *n*, obtained by selecting the Gaussian distribution minimizing the KLD. This approximation allows us to bound the complexity of the estimation process, which becomes independent of *N*. In the presence of multiple modes to estimate, the process is repeated until the *K* components have been detected (*K* is assumed to be known). After estimation of a ridge (the estimation of the IF associated with a component), the corresponding spectrogram vicinity values are set to zero before applying again the ridge extraction algorithm. This update of the TFR avoids multiple estimation of the same component. The overall procedure is detailed in Algorithm 1, where a backward correction is performed at line 9 after a first forward sequential estimation. This step has been added to avoid loss of performance in the first iterations due to possible slow convergence rate of the prior model.
**Algorithm 1:** Overall ridge extraction procedure using backward correction.  Input: TFR $S_{0}$, GRW mean $m_{0}$ and variance $\sigma_{0}^{2}$, Number of components *K*, *g*.  Ouput:
$\left[{\hat{m}}_{0},\cdots ,{\hat{m}}_{N-1}\right]$ for each component.   **for**
k=1,…,K
**do**   **for**
n=0,…,N−1
**do**    Compute $p(m_{n})$ by matching moments from Equtaion (Equation 10).    Compute the pseudo-posterior $p(m_{n}|s_{n})$ from Equtaion (Equation 8).    Perform MMSE estimation of ${\hat{m}}_{n}$.   **end for**   Repeat steps 5 to 7 iterating from n=N−1,…,0   Update the TFR by subtracting the *k*th ridge (TFR support set to 0).  **end for**

### 3.4. Alpha–Beta Divergence

The alternative choices to the KLD, namely the α-D and the β-D, for performing estimation of the IF are well adapted for regularizing the inferred solution. Although the α-D modifies the spread of the pseudoposterior distribution by controlling the cover of the empirical data mass function [21,27], the β-D allows it to compute a pseudoposterior distribution that is robust to outliers [21,28]. Here, outliers correspond to points unlikely to come from the data distribution due to strong noise contamination. Each divergence is associated with a hyperparameter, α or β, which controls the behavior of the variational objectives. Particular equivalences between objectives are obtained depending on the choice of α and β. For instance, both the α-D and the β-D are equivalent for considering the KLD when α→1 or β→0. Other equivalences are discussed in [21,28]. These two choices are motivated by the expected distribution of a noisy signal spectrogram. The distribution of $s_{n}$ can be modeled as a mixture of *K* Gaussian functions (because the STFT analysis window is a Gaussian function) whose means are the IFs of each component and weights are the relative amplitudes, plus a residual related to the noise. The postulated model in Equtaion (Equation 5) assumes the observation of a single sinusoidal component without any other additional source. Nonetheless, the presence of multiple components will precisely break the symmetry of the signal observations, resulting in skewed data distributions. Because the skewed Gaussian distributions are more affected on their tails, this motivates the use of the α-D in order to reduce the mass covering of the observation and to seek the mode associated with the distribution of each component. Moreover, such objective is desired when observing a signal with multiple components because the proposed algorithm aims to extract each ridge sequentially. In this work, we propose a new variational objective, based on the work in [21], aiming to enforce both the robustness of the estimate and the mode-seeking character of the approximated likelihood function. We consider the Alpha-Beta Divergence (ABD) [21,22], ∀α,β, such that α+β≠0,α≠0,β≠0 is expressed as
(11)$$\begin{array}{cc} D_{AB}^{\alpha ,\beta }(\hat{p}\left(s\right)\left|\right| & p\left(s|{\bar{m}}_{n}\right))=\frac{1}{\alpha (\alpha +\beta )}\int p{\left(s|{\bar{m}}_{n}\right)}^{\alpha +\beta }ds\\ \; & +\frac{1}{\beta (\alpha +\beta )}\int \hat{p}{\left(s\right)}^{\alpha +\beta }ds\\ \; & -\frac{1}{\alpha \beta }\int \hat{p}{\left(s\right)}^{\alpha }p{\left(s|{\bar{m}}_{n}\right)}^{\beta }ds. \end{array}$$

Note that the divergence in Equation (Equation 11) combines the objective of the α-, β- and γ-divergence as discussed in [22]. The γ-divergence [29] is another type of robust divergence that allows us, similar to the β-D, to reduce the influence of outliers during the approximation process. Both parameters in Equation (Equation 11) control the influence of the logarithm ratio in the likelihood term (see Equation (Equation 9)), through a weighting factor (resp. by deforming the logarithm factor) for α+β (resp. α) [22], as illustrated in Figure 1. The two parameters α,β involved in the divergence in Equation (Equation 11) give control on the resulting variational objective. Even though the robustness of the approximation is impacted by the value of α+β, modes can be highlighted through α. More precisely, the objective becomes robust to outliers and model mismatch when α+β>1, and it favors mode seeking when α<1. Nonetheless, α+β has to be chosen slightly above the threshold α+β=1 to ensure an efficient objective.

Although the estimation of the optimal divergence hyperparameters is out of the scope of this paper, we are interested by the property of the resulting estimators according to the hyperparameters choice. As discussed in [28], alternative objectives can be obtained by replacing the first term in Equation (Equation 8) by the CE of another divergence. As usually performed in variational inference, we drop the evidence term in order to construct the objective function. The resulting CE does not correspond anymore to the divergence, but to the ELBO to maximize for approximating the posterior distribution. The CE associated with the ABD in Equation (Equation 11) is
(12)$$\begin{array}{c} {\mathrm{CE}}_{AB}^{\alpha ,\beta }\left({\bar{m}}_{n}\right)=\frac{1}{\alpha \beta }\int \hat{p}{\left(s\right)}^{\alpha }p{\left(s|{\bar{m}}_{n}\right)}^{\beta }ds\\ -\frac{1}{\alpha (\alpha +\beta )}\int p{\left(s|{\bar{m}}_{n}\right)}^{\alpha +\beta }ds. \end{array}$$

This CE can then be incorporated into Equation (Equation 7) by using Equation (Equation 8). The pseudoposterior distribution used to infer estimates ${\hat{m}}_{n}$ gives [28]
(13)$$p\left({\bar{m}}_{n}|s_{n}\right)\propto e^{-M{\mathrm{CE}}_{AB}^{\alpha ,\beta }\left({\bar{m}}_{n}\right)}p\left({\bar{m}}_{n}\right).$$

### 3.5. Amplitude and Noise Estimation

Although the IFs associated with the mode of the MCS provides a useful knowledge for applications related to denoising, source separation or superresolution, it is not sufficient for reconstructing the signal components. Following the formulation in Equation (Equation 1), the amplitude associated with each component completes the information describing the MCS. In this section, we propose a simple method for estimating the amplitudes associated with each component in the presence of additive white Gaussian noise, assuming they have been previously estimated by using, for instance, the proposed PB strategy. In Algorithm 1, the amplitude associated with each component at each time index is estimated by using $|F_{y}^{h}\left[n,{\hat{m}}_{n}\right]|$. This is performed during the ridge-removal step (line 9 in Algorithm 1) to ensure that the whole energy of the frequency component is removed. Although such a step allows us to iterate over the *K* signal components through energy removal, the presence of spurious frequency content induced by noise avoids accurate amplitude modeling. Here, we propose an additional estimation step after the extraction of the ridge position to infer amplitude estimates from the spectrogram. More precisely, the method presented in Section 3.1 allows us to construct a mask, indicating the regions where the main energy associated with the informative content (the MCS) behaves. Hence, it also indicates the time-frequency points where the observation corresponds to noise. It has been shown [30] that the first statistical moments of the spectrogram of a white Gaussian noise are approximately constant. From this assumption, the noise statistics can be derived from the time-frequency points corresponding only to noise. The second statistical moment of a white circular Gaussian noise STFT rewrites
(14)$$\begin{array}{cc} \mathrm{Var}(F_{\epsilon }^{h}) & =E[\left(F_{\epsilon }^{h}-E\left[F_{\epsilon }^{h}\right]\right){\left(F_{\epsilon }^{h}-E\left[F_{\epsilon }^{h}\right]\right)}^{*}]\\ & =E[|F_{\epsilon }^{h}{|}^{2}]=E\left[S_{\epsilon }^{h}\right] \end{array},$$
giving an equivalence between variance of the STFT and the mean of the associated spectrogram, with $S_{\epsilon }^{h}\;=\;{\left|F_{\epsilon }^{h}\right|}^{2}$ the spectrogram of the white Gaussian noise signal. We propose to estimate $\mathrm{Var}(F_{\epsilon }^{h})$ by computing E[S] from regions mainly containing noise-related energy. The amplitude can then be estimated by using the spectrogram at the IFs after removal of the noise expected energy E[S].

## 4. Detection Algorithm

The simple observation model postulated in Equation (Equation 5) limits by its incompleteness the estimation performance of the amplitude and noise statistics, because both parameters are not involved in the observation model. Moreover, the proposed method is not adapted for signals whose components’ amplitude reach zero values in some time instants, because for each component the PB algorithm provides an IF estimate at each time instant. As a second part of this work, we thus propose a detection algorithm providing a more general estimation scheme in the presence of signal components behaving on a reduced subset of the time axis, or when IF estimation cannot be performed correctly due to the presence of important noise content [18,19]. Our new detection algorithm is based on a hypothesis test whose decision rule is formulated by using the marginal posterior distribution derived from an alternative observation model. More precisely, we develop in this section a test accounting for the presence (or absence) of a signal component at each time instant. The proposed detection algorithm requires the knowledge of ${\bar{m}}_{n}$, the amplitude *a*, and the expected noise level *b*. Hence, it is applied in the proposed framework after the estimation of these parameters.

### 4.1. Alternative Model

Let us consider a more general alternative observation model than the one proposed in Equation (Equation 5):(15)$$s_{n,m}|\left(w,{\bar{m}}_{n},u\right)∼u\left(1-w\right)g\left(m-{\bar{m}}_{n}\right)+w,$$
where $w=\frac{b}{a+b}\in \left[0,1\right]$ being the ratio of noise energy over the whole energy content, with *a* the positive amplitude and *b* the expected noise level, and where *u* is a boolean parameter indicating for the presence (u=1) or absence (u=0) of a ridge. If the observed signal corresponds only to noise (w=1), the observation model remains to be $p(s|a,b,{\bar{m}}_{n},u=0)\propto b$. Conversely, we have $p\left(s|a,b,{\bar{m}}_{n},u=1\right)\propto ag\left(s-{\bar{m}}_{n}\right)+b$ when a signal is present.

### 4.2. Prior Models

Prior distribution models are introduced here to account for the available prior knowledge on *a* and *b*. Indeed, a mixture prior distribution [31] is assigned to the amplitude parameter to model the presence or absence of a target
(16)$$p\left(a|u\right)=u\lambda_{a}e^{-\lambda_{a}a}+\left(1-u\right)\delta \left(a\right)$$
and an exponential prior distribution is associated with the noise expectation
(17)$$p\left(b\right)=\lambda_{b}e^{-\lambda_{b}b}.$$

Moreover, we assume independence between *a* and *b*, such that p(a,b)=p(a)p(b), giving the following joint prior distribution
(18)$$p\left(a,b|u,\Theta \right)=u\lambda_{a}\lambda_{b}e^{-\lambda_{a}a-\lambda_{b}b}+\left(1-u\right)\delta \left(a\right)\lambda_{b}e^{-\lambda_{b}b}$$
with $\Theta =(\lambda_{a},\lambda_{b})$. Equation (Equation 18) can then be formulated as a function of *w* and *b* by using a change of variable $a=b\left(\frac{1}{w}-1\right)$, such that
(19)$$\begin{array}{cc} p(w,b|u,\Theta ) & =u\lambda_{a}\lambda_{b}e^{-b\left(\lambda_{b}+\lambda_{a}\left(\frac{1}{w}-1\right)\right)}\\ \; & +\left(1-u\right)\delta \left(1-w\right)\lambda_{b}e^{-\lambda_{b}b}. \end{array}$$

Finally, we assign a Bernoulli prior model to the binary variable *u*, such that p(u=1)=ρ and p(u=0)=1−ρ.

### 4.3. Hypothesis Test

Similar to [32], the marginal posterior distribution is used to decide weather a target is present or not, such that
(20)$$p\left(u=1|s\right)\overset{H_{0}}{\underset{H_{1}}{≶}}p\left(u=0|s\right),$$
where the marginal posterior is defined such that
(21)$$p\left(u|s\right)=\int_{0}^{\infty }\int_{0}^{\infty }p\left(w,b,{\hat{m}}_{n},u|s\right)dbdw,$$
where the parameter ${\bar{m}}_{n}$ is replaced by an estimate ${\hat{m}}_{n}$ of the IF allowing us to avoid an integration over the ridge position. The marginal posterior p(u|s) can be computed by using the Bayes rule, such that
(22)$$p\left(u|s\right)\propto \int_{0}^{\infty }\;\;\;\;\int_{0}^{\infty }\;\;\;\;p\left(s|w,b,{\hat{m}}_{n},u\right)p\left(w,b|u,\Theta \right)p\left(u\right)dbdw.$$

### 4.4. Derivation

In this section, we derive p(u=0|s) and p(u=1|s) to decide if a ridge is present by using Equation (Equation 20). We begin with the simplest one:(23)$$\begin{array}{cc} p(u=0|s) & \propto \int_{0}^{\infty }b\lambda_{b}e^{-\lambda_{b}b}\left(1-\rho \right)db\\ \; & \propto \frac{\left(1-\rho \right)}{\lambda_{b}}. \end{array}$$

Similarly, the signal presence joint posterior probability writes (details available in Appendix C):(24)$$p\left(u=1,w|s\right)\propto \frac{g_{1}\left(w\right)\rho \lambda_{a}\lambda_{b}}{{\left[\left(\lambda_{b}+\lambda_{a}\left(\frac{1}{w}-1\right)\right)\right]}^{2}},$$
with
(25)$$g_{1}\left(w\right)=\prod_{m=1}^{M}\left[\left(\frac{1}{w}-1\right)g\left(m-{\hat{m}}_{n}\right)+1\right].$$

Note that no close form can be obtained for p(u=1|s), so we propose to numerically approximate it because the integral on *w* is not tractable. Once p(u=0|s) and p(u=1|s) have been computed by using, respectively, Equations (Equation 23) and (Equation 24), they are finally compared to make a decision following Equation (Equation 20).

### 4.5. Application to Multicomponent Signals

As previously discussed, the proposed detection step aims to enhance the time location of the reconstruction by truncating the components before synthesizing the TFR. Note that this approach can be plugged inside the PB algorithm to improve the efficiency of the GRW prior model. However, we restrict its use in this work to a postprocessing step for correcting the frequency band used for reconstruction. Because the PB algorithm estimates *K* frequency bands with a length of *N*, then each signal component is defined over the whole time axis. The wrong estimates are thus necessarily performed in time regions where all the components are not defined. The proposed estimation scheme cannot be extended directly to the study of MCS because the proposed models in both Equations (Equation 5) and (Equation 15) assume for the presence of only one component to describe the signal. Instead of reformulating the detection scheme developed in Section 4.1, we perform decision after the IF estimation by using Algorithm 1, where amplitude and noise first moments are estimated by using the method proposed in Section 3.5. These last two estimates provide useful knowledge for tuning $\lambda_{a}$ and $\lambda_{b}$ because good choices for these hyperparameters are $\lambda_{a}={\hat{a}}^{-1}$ and $\lambda_{b}={\hat{b}}^{-1}$, with $\hat{a}$ (resp. $\hat{b}$) standing for the estimated amplitude (resp. noise first moment), ensuring the mean of the exponential prior distributions to fit the expected parameters values. The detection is performed componentwise. More precisely, the knowledge of the IF allows us to separate the different components of the signal in the time-frequency plane in order to apply the detection step separately on each ridges. Temporary spectrograms are then generated (by applying for instance a thresholding mask on the original spectrogram) and used for the detection step. Once the detection arrays have been computed (one array per component) the related masks can be truncated to select, for each ridge, only the time-frequency instants containing signal information.

## 5. Robust Reconstruction

In this section, an alternative reconstruction methods is presented, where a new Non-Binary (NB) mask for band-pass reconstruction is introduced to circumvent the lack of reconstruction performance due to the presence of outliers in the vicinity of the estimated ridges. Indeed, MCS reconstruction is generally achieved through band-pass reconstruction, where only the associated informative time-frequency content is selected from the observed TFR [33]. This remains to restrict the integration region of Equation (Equation 35) to the vicinity of the IFs associated with the signal components. Such an approach improves the SNR of the signal in the presence of noise. Although this can significantly denoise the signal depending on the IF estimation performance, no particular attention is given to the remaining spurious noise in the vicinity of the signal component IF. When considering low SNR, an accurate estimation of the ridge positions indicating the signal frequency component does not allow for an efficient signal recovery due to the additional noise energy integrated in Equation (Equation 35). Here, we aim to replace the classical binary mask by another weighting filter to mitigate the noise content in the vicinity of the estimated IF. For this purpose, a truncated Gaussian function is used instead of a binary mask. More precisely, the reconstruction is performed by using the synthesis formula in Equation (Equation 35), by integrating a weighted TFR where the time-frequency points far from the components’ IF neighborhood are discarded.

First of all, we perform a preliminary analysis to motivate the use of such Gaussian filtering. For this purpose, we consider a single sinusoidal component signal whose IF is assumed to be known, merged with an additive white Gaussian noise. We then compare the performance of the Gaussian band-pass reconstruction filter with the classical binary one in terms of Reconstruction Quality Factor (RQF): $10\log_{10}\left(\frac{{\left||x|\right|}^{2}}{||x-\hat{x}{\left|\right|}^{2}}\right)$, where *x* (resp. $\hat{x}$) stands for the reference (resp. estimated) signal. While the spread of the Gaussian filter is controlled through its standard deviation (std), we use the three sigma rule of thumb to derive a binary filter with approximate width. Thus, for a given Gaussian std, we assign a binary filter with width equals to 6×std+1. After computing the RQFs obtained with both approaches for various SNR and filter spread, we compare their performance in Figure 2.

From Figure 2, we observe the difference between the RQFs computed with the Gaussian filter and the binary one. A positive value in Figure 2 thus indicates how much the RQF obtained by using a Gaussian reconstruction filter is better when compared to a binary filter for a given SNR and filter width. The black line delimits the transition between the two regions, where the Gaussian (resp. binary) performs better (resp. worse) than the other one. It can be observed from Figure 2 that the Gaussian filter provides better performance than the binary one. Indeed, performing classical band-pass filtering through a binary filter seems to be more efficient only for small width values.

This preliminary result is performed on a single component without frequency modulation rate [4,34] and aims to motivate the use of NB filtering reconstruction. We then compare the reconstruction performance of a sinusoidal component by using the proposed NB mask for various std and SNR. An experiment similar to the previous one is performed where the IF of the component is known. We thus construct a mask of width 2×10+1 frequency components centered around the ridge to filter the TFR. The results are displayed in Figure 3 from which two regions can be distinguished: above and below SNR ∼−7dB. For large SNRs, higher RQFs are obtained by considering broader Gaussian functions because the energy content is mainly informative. Conversely, an improvement can be observed in low SNR scenarios when highlighting the frequency values that are close to the estimated IF when performing the reconstruction.

## 6. Numerical Experiments

In this section, we assess (Matlab codes are freely available at https://codeocean.com/capsule/8693890/tree/v1, accessed on 5 November 2022) the methods proposed in this work for estimating the IF of signal components from their spectrogram, for detecting the temporal regions where the signal belongs and for improving the signal reconstruction performance when performing band-pass reconstruction in the presence of additional noise sources.

### 6.1. Synthetic Data Analysis

We aim to compare the performance of our algorithm to several state-of-the-art methods [11,20,33] on synthetic data for a first time. For this purpose, we consider a signal made of three components corresponding to a sinusoidal (C1), a linear chirp (C2) and a sinusoidally-FM component (C3) depicted in Figure 4 (from bottom to top).

In all the experiments conducted in this section, the STFTs are computed using M=500 and L=20. The temporal GRW prior model was initialized with $m_{0}=M/2$, $\sigma_{0}^{2}={\left(M/2\right)}^{2}/12$ and $\sigma_{rw}^{2}=2$. Moreover, an additive white Gaussian noise was used in all our simulations to model the presence of spurious content. In the remainder of this paper, we use the SNR to model the quality of the observations.

We first validate the use of the ABD by comparing its estimation performance to those of the α-D and β-D. Moreover, we also assess our method through a comparison with a recent Ridge Detector (RD) method [11]. First, we only consider the case of the component C1, and compare the IF estimation performance of the PB algorithm with different divergences by using the relative mean squared error
(26)$$\mathrm{RMSE}=\frac{1}{NM^{2}}{∥{\bar{m}}_{n}-{\hat{m}}_{n}∥}_{2}^{2},$$
where ${\bar{m}}_{n}$ (resp. ${\hat{m}}_{n}$) is the actual (resp. estimated) IF at the *n*th time instant. The estimation performance associated with the objective regions discussed in [21] are compared by using four different parameter pairs. More precisely, we consider the following set of parameters providing different objective characters when using the ABD: [α=0.4,β=0.4] (outliers focusing and mode seeking behavior), [α=0.2,β=1.5] (robustness and mode seeking behavior), [α=1.2,β=−0.4] (outliers focusing and mass-covering behavior), and [α=1.1,β=1.1] (robustness and mass-covering behavior). The estimation performance of the different methods according to the character of the variational approximations controlled by the divergence hyperparameters is assessed in Figure 5.

Although the performance of all the considered methods have a similar trend, their resilience to model mismatch can be assessed by comparing the SNR at which the RMSE increases. Note that the RMSE remains stable at high SNR for all the methods, and a loss of performance can be observed for SNRs in the range [−5,−10] dB. The less efficient estimations are obtained at high SNR by using the ABD with [α=1.2,β=−0.4], [α=1.1,β=1.1] and with the RD method. Those results show the inefficiency of improving the mass-covering character of the variational objective because even in the high SNR scenario it produces a flat approximate posterior distribution. Using [α=0.4,β=0.4] provides the best accuracy at high SNR by using the compared variational objectives of the proposed method while at low SNR we obtain estimation performance similar to the α-D and the β-D. Finally, the robust approach using [α=0.2,β=1.5] is mainly efficient at low SNR where it provides the best performances among the competing methods. It can be remarked that our most robust approach, [α=0.2,β=1.5], reaches better performance than when using the β-D with β∼1 [20].

In order to better understand the behavior of the different methods, we consider a single component signal made of a decreasing linear chirp (whose TFR is displayed in Figure 6 (top left)). In Figure 6 are displayed the estimation obtained by using the Brevdo method [33], the RD [11], and the proposed ABD approach. The two left-hand side columns correspond to results obtained when the signal is merged with an additive white Gaussian noise with a SNR of −5 dB, whereas the two right-hand side columns correspond to a SNR of −10 dB.

From Figure 6, it can be observed that in the presence of a moderate noise level (two left-hand side columns), the three compared methods manage to efficiently estimate the position of the ridge, even though the proposed approach provides a smoother estimate and more efficient tracking of the ridge. In the right-hand side case, the signal is more challenging to retrieve due to the important noise destroying the signal ridge. However, the proposed method achieves satisfactory estimation by enforcing the regularization in regions associated with important signal loss.

For a second time, we validate our method by comparing its reconstruction performance for the MCS depicted in Figure 4 with the Brevdo method [33] and the RD method [11]. For each method, we reconstruct a signal for each of the K=3 components C1, C2, and C3, by filtering the TFR by using a binary filter of width 2×10+1, centered around the IF of each component. The synthesis formula in Equation (Equation 35) is then applied on each band separately to extract the components. The use of the binary filter here is motivated by the difference in terms of performance of the Gaussian filter observed in Figure 3 according to the level of noise. It aims to ensure a fair comparison of the methods, without introducing an external bias due to to the prior lack of knowledge associated with the amount of noise in the observed signal. For a further comparison, we discard the approach working in the region enforcing the mass covering character of the variational approximation (α>1) because this objective does not provide satisfying results (see Figure 5).

The RQF obtained with the different approaches for a SNR of 10 dB with (resp. without) amplitude modulation are displayed in Table 1 (resp. in Table 2). More precisely, each component in Table 2 is assigned a distinct amplitude function. We respectively associate the components C1, C2, and C3 with the three amplitudes functions depicted in Figure 7.

In Table 1 (without amplitude modulation), it can be observed that the best performance is obtained by using parameter values satisfying α+β<1. Even though those values enforce the weights given to outliers, satisfying performance are still obtained due to the experimental conditions (moderate SNR). The proposed ABD methods perform better than the RD for all components, although the IF estimations associated with the sinusoidal component are similar when using both methods. Nonetheless, the robust methods (α+β>1) provide satisfying performance, except when both the robustness of the estimator is enforced and the mass of the observation is not well covered, avoiding efficiency of the variational approximation [22]. According to parameters choice, higher RQFs are obtained with our method in that particular case.

In Table 2 (with amplitude modulation), it can be observed that the proposed method is robust to variable amplitude. The performance of the proposed method does not significantly vary when compared to those in Table 1. Indeed, the method providing the best RQF for all components in average is the proposed ABD using α=0.4,β=0.4. Although the RD method collapses for the estimation of the third AM-FM component, it achieves slightly better reconstruction performance of C2. Even though the approaches enforcing the robustness of the estimator provides a satisfying performance for the linear chirp C2 and the FM component C3, they perform less efficiently to recover the IF associated with the sinusoid. This can be explained by the choice of the amplitude function of this component that should be the last component estimated by our algorithm due to its lower amplitude at the first time instants (see Figure 7). Considering α+β<1 allows us to achieve accurate estimation of the IF for the three considered components.

The computational time associated with the competing methods applied on the MCS depicted in Figure 4 is reported in Table 3 for different frequency resolutions. All the experiments have been computed by using Matlab R2021b with an Intel(R) Xeon(R) W-2123 CPU @ 3.60 GHz. Although the computational gain is implementation-dependent and could be improved via parallel programming, we notice that the proposed method as well as the RD [11] are more time consuming than the Brevdo method [33] when performing IF estimation for all frequency resolutions. Nonetheless, we remark that the complexity of the proposed method is similar but faster than [11]. The most computationally expensive step of the proposed algorithm is the computation of the pseudolikelihood, which has to be performed for each time instant. Nonetheless, this can be done as a preprocessing step because the pseudolikelihoods can be stored before running the Algorithm 1.

Now, we numerically assess the performance of the proposed detection approach. For that purpose, we first consider a truncated version of the component C1, displayed in Figure 8 (left), such that it is assigned a binary amplitude taking values in [0,1]. The spectrogram of this signal is displayed in Figure 8 (right).

We assess the performance of the detection method through a comparison with [20,33] using the following Mean Absolute Error (MAE) defined as
(27)$$\mathrm{MAE}=\frac{1}{N}\sum_{n=0}^{N-1}\left|{\bar{d}}_{n}-{\hat{d}}_{n}\right|,$$
where ${\bar{d}}_{n}$ (resp. ${\hat{d}}_{n}$) is the actual (resp. estimated) presence array at the *n*-th time instant. Our evaluation also considers an Oracle estimator, which provides a binary mask computed by using the threshold ground truth isolated component signal. This estimator aims to show the influence of the parameter $\lambda_{a}$ because the knowledge of the ridge position allows us to estimate the signal amplitude more efficiently. This comparison highlights the influence of the IF estimation performance during the detection process. Moreover, it is preferable to significantly increase the GRW parameter $\sigma_{rw}$ when performing detection, because ridges are not necessarily observed in the first time instants. Thus, the PB algorithm will estimate a ridge position around local maxima corresponding to noise and propagate this information to the following time instants. By increasing $\sigma_{rw}$, we ensure that wrong prior models, due to the absence of observed signal, will not avoid efficiently estimating the IF as long as the pseudolikelihood is sufficiently informative. For that reason we set $\sigma_{rw}=10$ for the experiments conduced in this section. The resulting MAE obtained with the different competing methods are displayed in Figure 9.

It can be observed from the results in Figure 9 that the Oracle accurately detects the signal for SNR >−15 dB, showing the importance of the estimation performance of the parameters required to achieve the detection. The IF estimation performance enhances the detector accuracy through p(u=1,w|s) in Equation (Equation 24), and improves the estimation performance of the amplitude and noise used in the decision rule. The other approaches perform satisfactorily for a SNR ≥−5 dB. The estimation performance of the variational objectives is similar to previous results, because the robust methods provide better results at low SNR. It can be remarked that when α+β>1, increasing the value of α+β improves the performance of the detection algorithm. Moreover, reducing the mode-seeking character provides better performance than enforcing the robustness of the objective (better results are obtained by using [α=0.7,β=1.2] than with [α=0.2,β=1.5]). When selecting α+β>1, a better detection is performed by using slight mode seeking objective. When highlighting the importance of outliers in the IF estimation, focusing on the distribution mode can avoid proper ridge following, corrupting the detection process.

We finally assess the signal reconstruction performance of the alternative method based on the SST and reconstruction by using the NB mask discussed in Section 5. For the latter case, we compare the reconstruction performance obtained with different masks associated with distinct std. Even though the following experiments present the performance obtained by using [α=0.4,β=0.4], we have empirically observed during our experiments that similar results are obtained when working in the regions given by α<1 and α+β<1. For comparison purposes, we also assess the alternative reconstruction performance on synchrosqueezed signals. We are thus interested by the efficiency of the SST-based approaches [10,35] that allow us to highlight the information related to the signal by improving its time-frequency localization. There is no restriction to apply the proposed PB method on such a modified TFR as long as the distribution associated with the data in Equation (Equation 13) is updated accordingly. Because no analytic formulation of the data distribution can be derived from such a representation, we empirically estimate the parameters of the Gaussian analysis window. The resulting distribution is proportional to a Gaussian function with std = 0.5. The RQF of each component of the MCS in Figure 4 are respectively displayed for components C1, C2, and C3 in Figure 10 (top left, top right, bottom).

We observed from Figure 10 (top left, top right, and bottom), that although our method provides performance similar to the Brevdo approach, the proposed alternative reconstruction methods improves the reconstruction of each component for low SNRs. Indeed, applying our algorithm on the SST instead of the STFT enhances the reconstruction through a higher RQF for SNR <−7 dB. For SNR <−5 dB, the best reconstruction in term of RQF among the compared method is given by the proposed NB with low std. However, such a choice reduces the quality of the reconstruction for high SNRs because it tends to reduce the amount of information used to reconstruct the signal components. Similar to what is observed in Figure 3, considering smaller std has a denoising effect and thus improves the quality of the reconstructed signal at low SNRs. Conversely, considering broader Gaussian provides results that are close to the original method and thus increases the sensitivity of the reconstruction to the presence of noise. The proposed approach improves the RQF by reducing the energy used to synthesize the signal producing a slight loss of information.

### 6.2. Real-World Data

We consider first a signal corresponding to sounds emitted by a piping queen bee investigated in [36] which has been downsampled to 16 kHz and truncated to obtain the result depicted in Figure 11, where the estimated IF of each mode is superimposed with different colors. Despite the exact IF ground truth is not available, the ridges corresponding to the piping signal components are quite visible and match with our estimation.

The K=5 estimated modes are obtained by using a STFT analysis window spread L=20 and with our proposed ABD using α=0.2 and β=0.6 (postprocessed by using our detection algorithm) are shown in Figure 11. This figure illustrates the performance of our method, where the modes are well-retrieved on each side of the TFR. Moreover, this clearly shows the interest of the detector for avoiding the detection of spurious components in the [0.13,0.26]s range. Secondly, we apply our proposed approach on a 2.5-ms-long echolocation pulse signal emitted by a Eptesicus Fuscus bat (https://www.ece.rice.edu/dsp/software/bat.shtml, accessed on 5 November 2022), sampled at $T_{s}=7\;\mu s$.

The estimated K=3 signal modes obtained with our PB algorithm using α=0.4 and β=0.7 are shown in Figure 12, with (right) and without (left) application of the detection method. From Figure 12, we can distinguish the ability of the method to retrieve the modes of the signal when the amplitude and frequency of each components significantly vary over time. The result obtained after application of the detection algorithm is depicted in Figure 12. In Figure 12 (right), we observe the ability of the method to efficiently decide for the presence or absence of the ridges even when the observed data distribution is strongly modulated.

## 7. Conclusions

In this paper, we have proposed a new pseudo-Bayesian method designed for ridge detection, mode disentangling, and reconstruction in the presence of noise. The proposed approach combines three distinct problems which are often separately addressed in the literature.
1.A new, robust, instantaneous frequency estimator has been proposed to perform estimation of the ridge position in the time-frequency plane accounting for the presence of spurious additional noise. The simple postulated observation model allows us to quickly infer estimates by sequentially extracting the instantaneous frequency associated with each component of the signal. The new variational objective that is proposed in this work controls together the balance robustness/efficiency and mode seeking/mass covering of the estimator.2.An algorithm to perform signal detection in order to postprocess the instantaneous frequency estimation is based on a hypothesis test requiring amplitude and frequency noise-expectation estimates. We showed the ability of the proposed detection method to efficiently estimate the time instants when the signal is active, as well as the importance of the amplitude estimation performance for achieving a satisfying signal reconstruction.3.We alleviate issues encountered when performing signal reconstruction from noisy frequency bands of the STFT. We finally present in this work two different denoising reconstruction approaches, involving, respectively, a simple extension of the proposed algorithm to apply on the signal SST, and the use of a nonbinary mask.

The experiments conducted in this work show that the proposed variational objective can provide better estimation results than the compared state-of-the-art methods, in particular the RD method, which was recently introduced in [11]. Future work includes the optimization of the divergence hyperparameters selection and their application in new real-world sensor-based applications. Moreover, a further investigation of other nonbinary masks for signal reconstruction could also be full of interest in specific denoising scenarios.

## Figures and Tables

**Figure 1 sensors-23-00085-f001:**
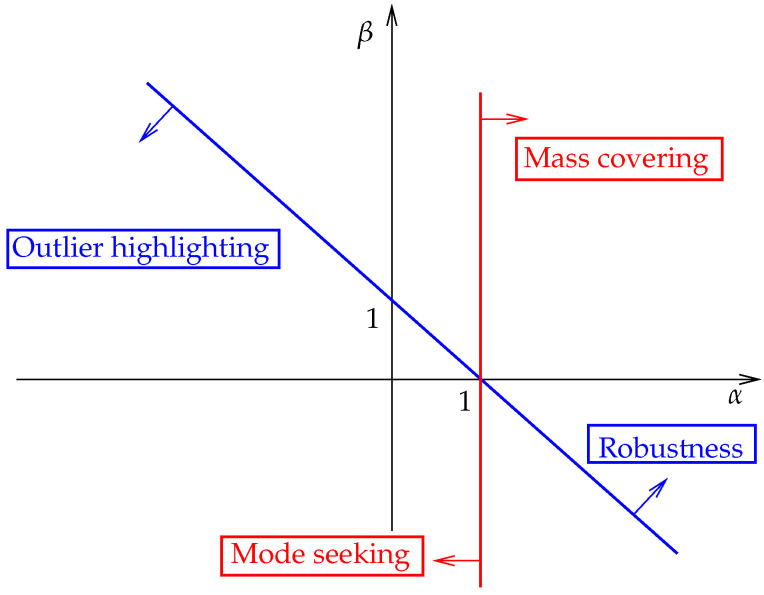
Character of the variational objective obtained by varying the ABD hyperparameters α and β.

**Figure 2 sensors-23-00085-f002:**
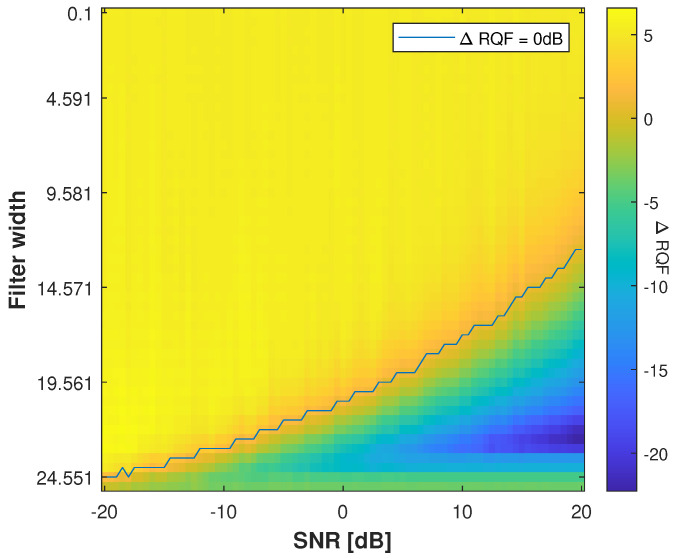
RQF difference between the Gaussian and binary filter. Each pixel is associated with a distinct SNR and reconstruction filter bandwidth. A positive values informs on the denoising performance of the Gaussian filter over that of the binary one. Conversely, a negative value means the binary filter reaches higher RQF. The results are averaged over 100 realizations of noise. The black line delimits the positive and negative regions.

**Figure 3 sensors-23-00085-f003:**
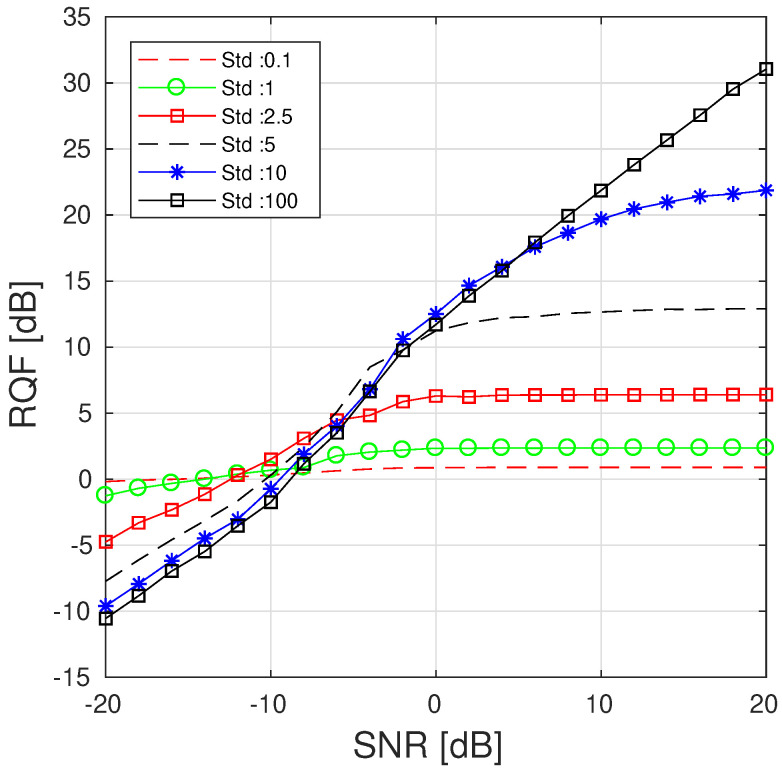
RQFs obtained by using the Gaussian filter approach for various SNR and std for L=20. The results are averaged over 100 realizations of noise.

**Figure 4 sensors-23-00085-f004:**
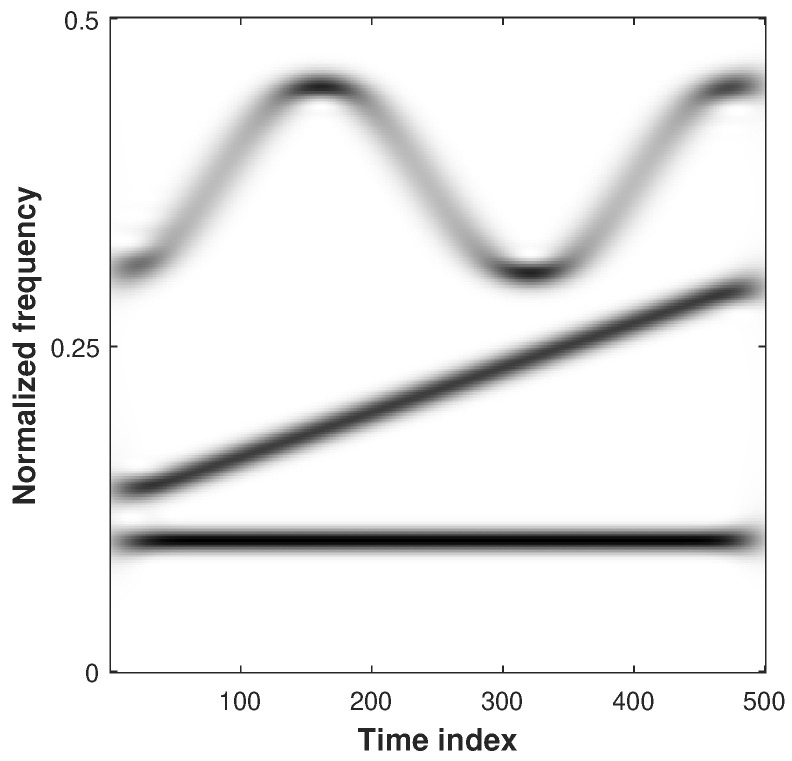
Spectrogram of the analyzed multicomponent signal.

**Figure 5 sensors-23-00085-f005:**
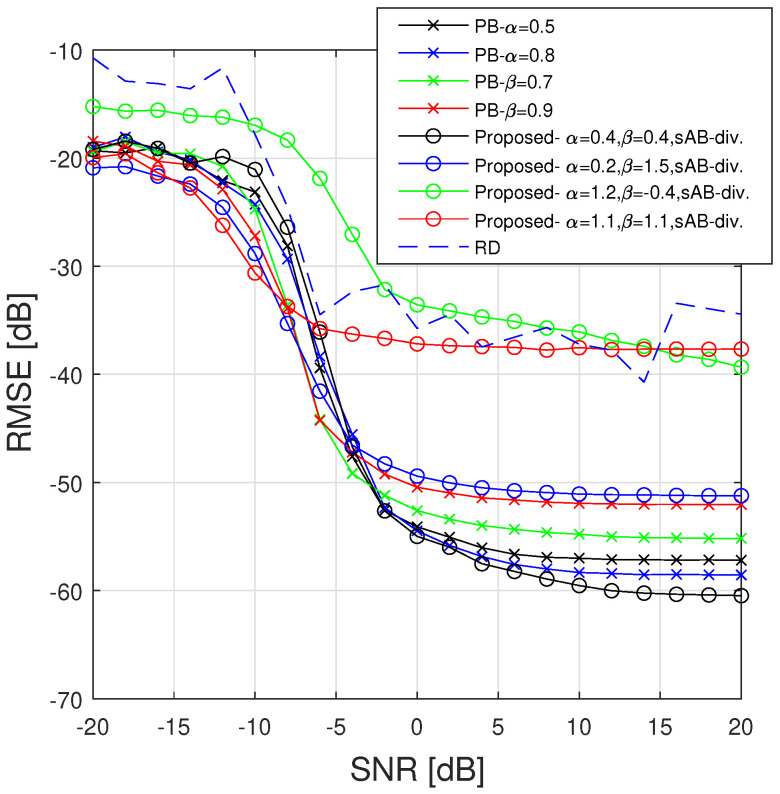
RMSE (in dB) of the IF (averaged over 100 realizations of noise) obtained with the different competing methods for the component C1 [23,24].

**Figure 6 sensors-23-00085-f006:**
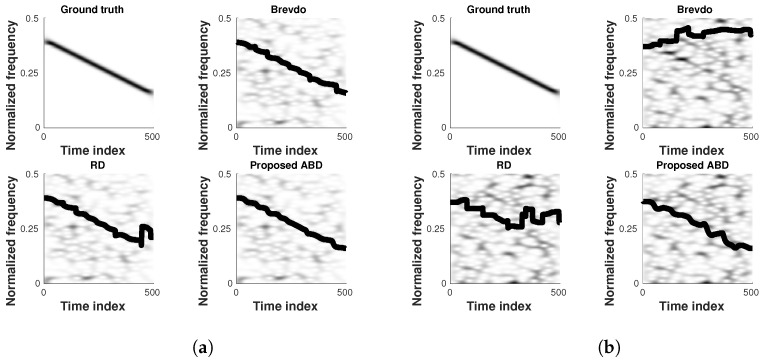
Estimation example of a single component signal in the time-frequency plane using the competing methods respectively at SNR = −5 dB (**a**) and SNR = −10 dB (**b**).

**Figure 7 sensors-23-00085-f007:**
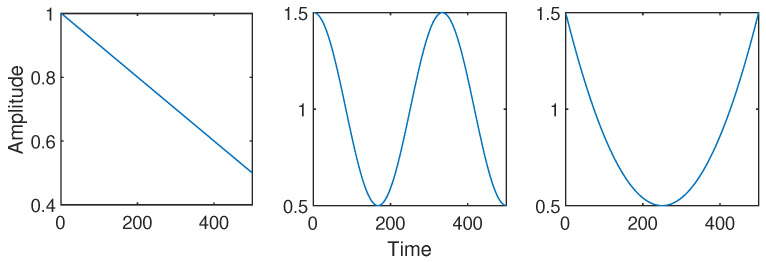
Amplitudes associated with components C1 (**left**), C2 (**middle**), and C3 (**right**).

**Figure 8 sensors-23-00085-f008:**
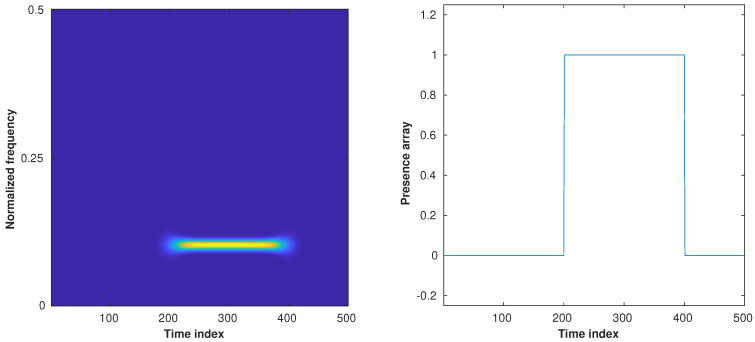
**Left**: Spectrogram of a noiseless monocomponent signal whose frequency is not null in time indices [200,400]. **Right**: Ground truth binary detection indicating where the signal amplitude is non-null.

**Figure 9 sensors-23-00085-f009:**
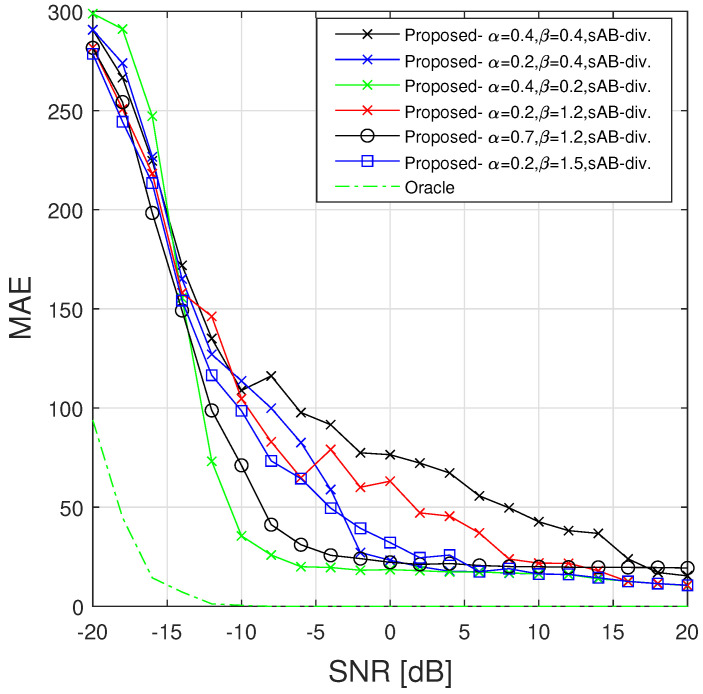
MAE between estimated detection array and ground truth for a linear chirp (truncated middle component in Figure 4) by using L=20 (obtained with 100 realizations of noise).

**Figure 10 sensors-23-00085-f010:**
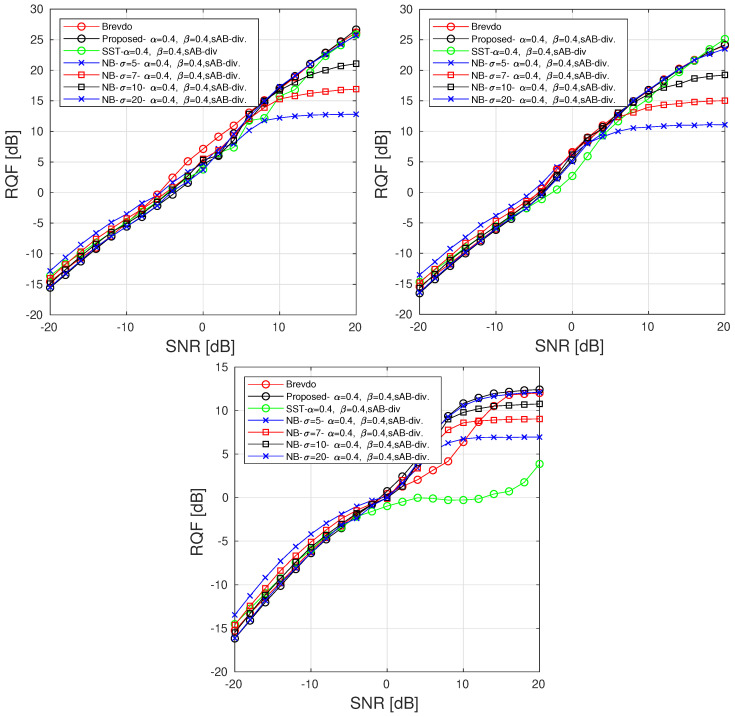
RQF obtained for each component in Figure 4 using compared methods for various SNRs with L=20. The results are averaged over 100 realizations of noise [5].

**Figure 11 sensors-23-00085-f011:**
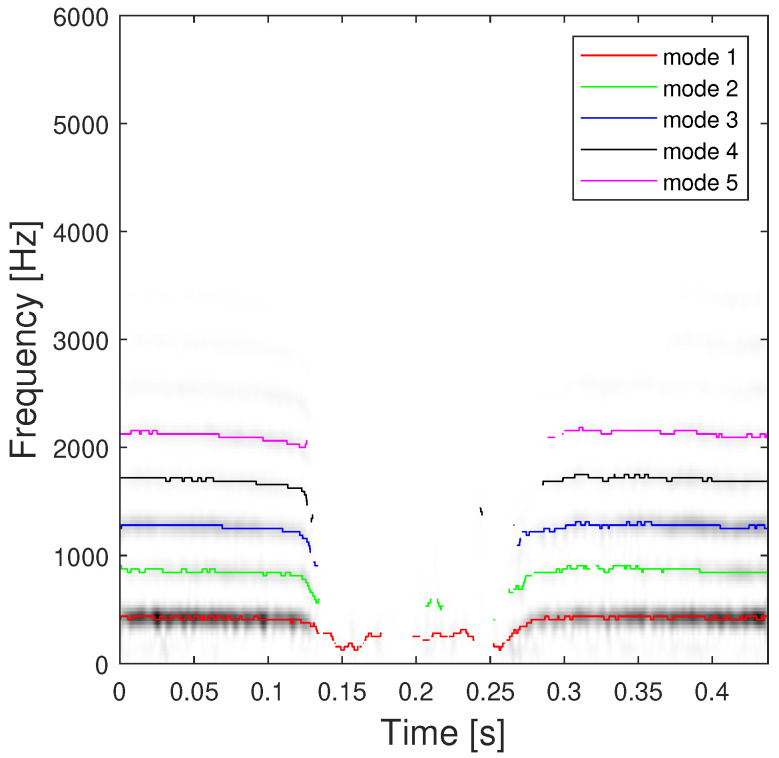
Estimation of the first K=5 signal components of the piping data using the proposed ABD method.

**Figure 12 sensors-23-00085-f012:**
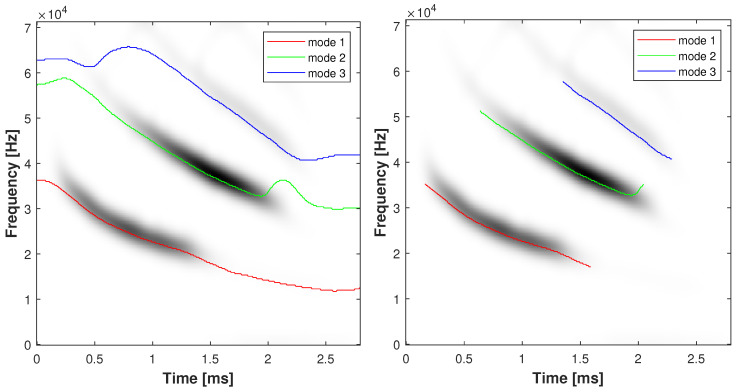
Estimation of K=3 signal components of the bat record signal by using the proposed ABD method with α=0.4,β=0.7 with (**right**) and without (**left**) performing the detection.

**Table 1 sensors-23-00085-t001:** RQF of each reconstructed components (averaged over 100 realizations of noise) for the different competing approaches for a SNR = 10 dB without AM. In second rows are displayed, for each case, the std of the estimators.

	C1	C2	C3	Average
Brevdo [33]	15.86±0.84	16.60±0.84	6.52±2.12	12.70±1.4
RD [11]	16.22±7.26	12.28±7.14	5.05±7.92	11.18±7.45
ABD, α=0.4,β=0.4	17.12±1.81	16.63±0.81	11.04±0.81	14.31±0.63
ABD, α=0.2,β=0.4	17.07±0.79	16.64±0.78	10.89±0.62	14.26±0.74
ABD, α=0.4,β=0.2	16.93±0.89	16.52±0.76	10.90±0.57	14.28±0.75
ABD, α=0.2,β=1.2	14.96±6.74	16.78±0.80	9.33±0.64	13.35±3.93
ABD, α=0.7,β=1.2	11.62±4.84	16.37±0.72	9.51±0.33	12.20±8.03
ABD, α=0.2,β=1.5	5.52±8.90	16.28±2.43	8.64±0.24	9.85±2.83

**Table 2 sensors-23-00085-t002:** RQF of each reconstructed components (averaged over 100 realizations of noise) for the different competing approaches for a SNR = 10 dB with AM. In second rows are displayed, for each case, the std of the estimators.

	C1	C2	C3	Average
Brevdo [33]	15.86±0.84	16.60±0.84	6.52±2.12	12.70±1.4
RD [11]	16.22±7.26	12.28±7.14	5.05±7.92	11.18±7.45
ABD, α=0.4,β=0.4	17.12±1.81	16.63±0.81	11.04±0.81	14.31±0.63
ABD, α=0.2,β=0.4	17.07±0.79	16.64±0.78	10.89±0.62	14.26±0.74
ABD, α=0.4,β=0.2	16.93±0.89	16.52±0.76	10.90±0.57	14.28±0.75
ABD, α=0.2,β=1.2	14.96±6.74	16.78±0.80	9.33±0.64	13.35±3.93
ABD, α=0.7,β=1.2	11.62±4.84	16.37±0.72	9.51±0.33	12.20±8.03
ABD, α=0.2,β=1.5	5.52±8.90	16.28±2.43	8.64±0.24	9.85±2.83

**Table 3 sensors-23-00085-t003:** Computational cost of the competing approaches for synthetic data analysis. The results have been obtained by averaging over 50 realizations.

	M=500	M=1000	M=2000	M=5000
Brevdo [33]	0.05 s	0.07 s	0.13 s	0.36 s
RD [11]	1.52 s	2.54 s	4.47 s	12.29 s
ABD	0.28 s	0.49 s	1.20 s	7.15 s

## Data Availability

All the experiments performed in this work have been computed by using Matlab R2021b with an Intel(R) Xeon(R) W-2123 CPU @ 3.60GHz. The codes are all available on CodeOcean at https://codeocean.com/capsule/8693890/tree/v1 (accessed on 5 November 2022).

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
