# Peer review of "Pseudo-Bayesian Approach for Robust Mode Detection and Extraction Based on the STFT"

_sensors, 2022, doi:10.3390/s23010085_

Round 1

Reviewer 1 Report

Dear authors,

Thank you very much for this interesting manuscript. 

Please find below some general comments:

The manuscript lacks a proper discussion and conclusion. There is only a summary at the end (section 8) that is referred to as "Discussion".

The manuscript also lacks a clear motivation of your research. Even though you apply your approach to real-world data, it is not clear how signal decomposition can assist the analysis of this data and how the presented approach improves this assistance. 

The structure of the manuscript is confusing. For instance, section 7.4 Real-world data is a sub-section of section 7. Numerical experiments where synthetic data are introduced and used for the tests in 7.1 to 7.3. Additionally, the sub-subsections in section 4 should be promoted to subsections and subsection 4.1 be removed.

It is not clear how the Pseudo-Bayesian Approach for Robust Multi-Ridge Detection presented in this manuscript compares to the approach in [28]. What is new, what is the same. To me both seem very similar. For instance, Algorithm 1 is very similar (or the same) to Algorithm 1 in [28] without proper reference. The only difference I can see is the use of alpha-beta-D in contrast to beta-D in [28]. Therefore, it seems that the manuscript is not revealing novel scientific output.

In Section 7 figures of the reconstructed signals and individual components in the time and time-frequency domain would help to better understand the performance of the approach.

Author Response

Comment 1.1
The manuscript lacks a proper discussion and conclusion. There is only a summary at the end (section 8) that is referred to as "Discussion".

Response:
Thanks for this comment. The Discussion section that appeared at the end of the paper has been revised. It is now treated and referred to as a Conclusion. Moreover, we reorganized this section to enhance its readability, and slightly modified its content to emphasize its concluding purpose.

Comment 1.2
The manuscript also lacks a clear motivation of your research. Even though you apply your approach to real-world data, it is not clear how signal decomposition can assist the analysis of this data and how the presented approach improves this assistance.

Response:
Thanks for this comment. We modified the introduction of the paper to make
more clear the link between signal analysis and the estimation performed in the
paper. This discussion was indeed missing in the original version, leading to a
lack of clarity of the motivation behind this work. We hope this new version will
convince the reviewers about the usefulness of this work for addressing a large
variety of signal processing problems involving sensors.

Comment 1.3
The structure of the manuscript is confusing. For instance, section 7.4 Real-world data is a sub-section of section 7. Numerical experiments where synthetic data are introduced and used for the tests in 7.1 to 7.3. Additionally, the sub-subsections in section 4 should be promoted to subsections and subsection 4.1 be removed.

Response:
Thanks for this comment. The structure of the paper has been modified and
improved according to the reviewer comments. Now, Section 4 contains subsections where the previous subsection 4.1 is removed. Section 7 is now structured into two subsections: synthetic and real-world data. According to the Editor’s comments, we also removed the previous Section 2 that introduces the notations and definition. This section has then be moved both to Section 3 (which is Section 2 now) and to the Appendix.

Comment 1.4
It is not clear how the Pseudo-Bayesian Approach for Robust Multi-Ridge Detection presented in this manuscript compares to the approach in [28]. What is new, what is the same. To me both seem very similar. For instance, Algo. 1 is very similar (or the same) to Algorithm 1 in [28] without proper reference. The only difference I can see is the use of alpha-beta-D in contrast to beta-D in [28]. Therefore, it seems that the manuscript is not revealing novel scientific output.

Response:
Thanks for this comment. The differences and contributions in comparison to
our previous work [28] are numerous. First, we would like to emphasize that the proposed estimation algorithm is not the only contribution of this work, which also proposes a novel detection method for estimating the time region over which the signal components are defined, as well as a new non-binary mask aiming to improve the signal reconstruction performance when resorting to band-pass synthesis after ridge estimation.
Regarding the first contribution of this paper, namely the pseudo-Bayesian algo-
rithm, we expect in this work to conclude what is presented in [28]. This is done by generalizing the variational objective using a unique alternative divergence, which has the advantage of providing better performance, to be more user friendly (a single divergence and a better understanding of the divergence hyperparameter)
and to provide a better control on the estimation process. While this associated
divergence choice is not trivial (compared to the two simple divergences (alpha- and beta-) used as a proof of concept in [28]), it provides better results than previous works.
From a theoretical point of view, the contribution of this work can be summarized in a Bayesian context, to a redefinition of the observation model. Indeed, even though we are not working directly on the observation model, it is indirectly modified when the likelihood term is replaced by a novel alternative divergence. In practice, this new divergence choice is equivalent to modify the "cost function" associated with the estimation problem considered. Similarly to optimization approaches, changes in the estimations strategy can involve important modifications of the expected solution. To keep the analogy with optimization approaches,
we reformulate the data term in this work while the optimization does not differs from [28]. The algorithm presented in Algo. 1 is similar to that of [28], since the estimation is performed similarly, even though the variational objective is utterly different. For instance, two convex optimization approaches based on different models are not identical because they both rely on gradient ascent. Last but not least, we would like to highlight the theoretical output made on the model formulation.
Indeed, we strengthen in this work the theoretical basis of our observation model through the derivations made in Appendix A. While the same observation model was used in [28], it was only assumed to approximately match the observations, while we prove here that our model is exact when observing a single sinusoidal signal component(noiseless case). While this might seems specific, this is a sufficient motivation for the assumed observation model we use.

Reviewer 2 Report

The authors propose a new approach for Robust Mode Detection and Extraction based on the STFT. The presented idea might be valuable, but the description of the algorithm need to be enhanced. Some of the references cited are of long duration, lack of research significance, and the latest literature is a little less. The paper is quite interesting even if it is advisable to better describe the aims and simulation results of the tests carried out.

Author Response

Comment 2.1
The authors propose a new approach for Robust Mode Detection and Extraction based on the STFT. The presented idea might be valuable, but the description of the algorithm need to be enhanced.

Response:
Thanks for this comment. The structure of the manuscript has now been revised
according to the Editor and Reviewer 1 comments to emphasize the contribution, to enhance the readability of the results and to clarify the motivation behind this work. We hope this will now convince you about the novelty and usefulness of the proposed approach.

Comment 2.2
Some of the references cited are of long duration, lack of research significance, and the latest literature is a little less.

Response:
Thanks for this comment. For the sake of clarity, we have entirely revised all
the references of the paper to only select the more relevant and recent ones. As a result, the length of the bibliography section has significantly decreased.

Comment 2.3
The paper is quite interesting even if it is advisable to better describe the aims and simulation results of the tests carried out.

Response:
Thanks for this comment. Since our proposed method can be used in a large
variety of signal analysis/sensing problems, we first validate our approach in a
simulated scenario involving sinusoidal components merged with a white Gaussian noise (with a comparison with the state-of-the-art). Second, we present two distinct examples of the proposed approach (combining ridge estimation and signal detection) applied to real-world signals: 1)a piping signal recorded from a beehive by a microphone (sampling rate 16kHz), 2)an echolocation pulse signal emitted by a bat (sampling rate about 142.86 kHz).

Round 2

Reviewer 1 Report

Dear Authors,

thanks a lot for considering my comments to the first submission and your thorough clarification especially regarding Comment 1.4. 

Since the manuscript has been fundamentally improved, I would be happy to see the manuscript published in the present form after some final editing of English language and style.

Best regards,